# Protective Effect of *Colla corii asini* against Lung Injuries Induced by Intratracheal Instillation of Artificial Fine Particles in Rats

**DOI:** 10.3390/ijms20010055

**Published:** 2018-12-23

**Authors:** Tiantian Liu, Piaopiao Zhang, Yahao Ling, Guang Hu, Jianjun Gu, Hong Yang, Jinfeng Wei, Aiping Wang, Hongtao Jin

**Affiliations:** 1State Key Laboratory of Bioactive Substance and Function of Natural Medicines, Institute of Materia Medica, Chinese Academy of Medical Sciences & Peking Union Medical College, Beijing 100050, China; liutiantian@imm.ac.cn; 2New Drug Safety Evaluation Center, Institute of Materia Medica, Chinese Academy of Medical, Sciences & Peking Union Medical College, Beijing 100050, China; purepiao123@gmail.com (P.Z.); lingyahao1990@163.com (Y.L.); huguang@imm.ac.cn (G.H.); weijinfeng@imm.ac.cn (J.W.); wangaiping@imm.ac.cn (A.W.); 3National Engineering Research Center for Gelatin-based Traditional Chinese Medicine, Shandong 252299, China; gujj@dongeejiao.com; 4Experimental Research Center, China Academy of Chinese Medical Sciences, Beijing 100700, China; yanghong@merc.ac.cn; 5Beijing Union-Genius Pharmaceutical Technology Co., Ltd., Beijing 100176, China

**Keywords:** *Colla corii asini*, artificial fine particles, intratracheal instillation, lung injury, metabolomics

## Abstract

Environmental issues pose huge threats to public health, particularly the damage caused by fine particulate matter (PM_2.5_). However, the mechanisms of injury require further investigation and medical materials that can protect the lungs from PM_2.5_ are needed. We have found that *Colla corii asini*, a traditional Chinese medicine that has long been used to treat various ailments, is a good candidate to serve this purpose. To understand the mechanisms of PM_2.5_-induced lung toxicity and the protective effects of *Colla corii asini*, we established a rat model of lung injury via intratracheal instillation of artificial PM_2.5_ (aPM_2.5_). Our results demonstrated that *Colla corii asini* significantly protected against lung function decline and pathologic changes. Inflammation was ameliorated by suppression of *Arg-1* to adjust the disturbed metabolic pathways induced by aPM_2.5_, such as arginine and nitrogen metabolism and aminoacyl-tRNA biosynthesis, for 11 weeks. Our work found that metabolomics was a useful tool that contributed to further understanding of PM_2.5_-induced respiratory system damage and provided useful information for further pharmacological research on *Colla corii asini*, which may be valuable for therapeutic intervention.

## 1. Introduction

Air pollution, particularly fine particles (particles with a mass median aerodynamic diameter of less than 2.5 μm, PM_2.5_), has become a major environmental issue affecting public health globally [1,2,3]. Although air quality-control policies have been implemented by the Chinese government, the levels of PM_2.5_ in China remain substantially higher than those in developed countries and have become a significant threat to people living on the mainland [4,5,6]. Studies suggest that short- and long-term exposure to PM_2.5_ are linked to a range of negative health outcomes, including respiratory disorders [7], cardiovascular and cerebrovascular diseases [8] and decreased life expectancy [9,10].

*Colla corii asini* (E’jiao, donkey-hide gelatin), a traditional Chinese medicinal preparation that has been used for more than 2000 years, is the solid glue prepared from the skin of Equus asinus by boiling and concentration. The main components are amino acids and microelements, as well as small molecular weight hydrolysates of collagen proteins that are included in the medical material. It has been reported to possess a variety of clinical functions in terms of hematopoiesis, enhancement of cellular immunity and radio-protection [11,12,13,14]. Recent reports have shown that *Colla corii asini* has anti-inflammation properties and can stimulate both specific and non-specific immune function in hypoimmune mice. Zhang showed that *Colla corii asini* reversed the imbalance of Th17/Treg subsets in mice with airway inflammation [15,16]. Our previous study showed that *Colla corii asini* prevented oxidative damage caused by aPM_2.5_ [17]. These results collectively suggest that *Colla corii asini* may be a useful therapeutic agent to protect against pulmonary impairment.

The mechanisms underlying PM_2.5_-induced lung toxicity/pulmonary impairment still need further study. To explore this issue, artificial PM_2.5_ particles (aPM_2.5_) were created based on the physical and chemical properties of PM_2.5_ particles collected in Beijing [18]. We investigated the protective effects and mechanism of action of *Colla corii asini* in an 11-week study of respiratory system injury induced by intratracheal instillation of aPM_2.5_ in rats.

## 2. Results

### 2.1. Colla corii asini Protects Rats from aPM_2.5_-Induced Lung Function Changes

The rats were exposed to aPM_2.5_ (30 mg/kg, 1 mL/kg body weight) via intratracheal instillation except for the controls, while rats in the aPM_2.5_ + *Colla corii asini* groups received *Colla corii asini* daily by oral gavage for 11 weeks at doses of 1, 2 and 5 g/kg body weight. Respiratory function of all animals was measured weekly throughout the treatment for 11 weeks. As shown in Figure 1A–D, compared with those in the control group, the tidal volume, expiratory volume, minute ventilation volume and expiratory flow at 50% tidal volume of the aPM_2.5_ group were significantly increased between 8 and 11 weeks. Treatment with *Colla corii asini* significantly suppressed these volume parameters (* *p* < 0.05 or ** *p* < 0.01). The inspiratory time, end-inspiratory time and respiratory rate of aPM_2.5_-treated rats decreased (Figure 1E,F,I) with respect to the control group but these parameters were increased in the aPM_2.5_ + *Colla corii asini* groups (* *p* < 0.05 or ** *p* < 0.01). Airway resistance was increased after intratracheal instillation of aPM_2.5_ between 4 and 11 weeks, as evidenced by the change in flow index of the ventilation function, including increases in peak inspiratory and expiratory flow. After *Colla corii asini* treatment, these indices significantly rebounded (* *p* < 0.05 or ** *p* < 0.01), with values similar to those of controls (Figure 1G,H). Intratracheal instillation of aPM_2.5_ induced a time-dependent reduction of lung function. Treatment of aPM_2.5_-exposed rats with *Colla corii asini* (1, 2, or 5 g/kg, oral) for 11 weeks provided a significant protective effect against aPM_2.5_-induced lung function changes.

### 2.2. Effect of Colla corii asini on aPM_2.5_-Induced Alteration of T Lymphocyte Subsets

The T lymphocyte profiles (Table 1) were not significantly different between the control and aPM_2.5_ groups in the fresh whole blood of rats of 11 weeks. However, *Colla corii asini* decreased CD8^+^ level in the presence of aPM_2.5_. Compared with the aPM_2.5_ group, the CD4^+^/CD8^+^ ratio increased in the *Colla corii asini* + aPM_2.5_ groups.

### 2.3. Effect of Colla corii asini on Cell Counts and Percentages

Indicative of lung inflammation, significant increases in total white blood cell (WBC), lymphocyte (LYM), monocyte (MON), neutrophil (NEUT) and eosinophil (EOS) counts in BALF week 11 samples were observed in the aPM_2.5_-exposed rats compared with the controls (Figure 2A, ^##^
*p* < 0.01). The most pronounced increases were in the LYM% and BAS% counts (Figure 2B, ^#^
*p* < 0.05). *Colla corii asini* treatment reversed these changes compared to the aPM_2.5_ group (* *p* < 0.05 or ** *p*< 0.01).

### 2.4. Effect of Colla corii asini on Protein and mRNA Expression of TNF-α, IL-1β and IL-10

As shown in Figure 3A, aPM_2.5_ caused increased production of pro-inflammatory cytokines, including TNF-α and IL-1β (^##^
*p* < 0.01) and a decrease of anti-inflammatory IL-10 (^#^
*p* < 0.05) in BALF compared to the control rats. In particular, the high-dose of *Colla corii asini* decreased the protein level of TNF-α (* *p* < 0.05), while IL-1β decreased as the *Colla corii asini* dose was increased (* *p* < 0.05 or ** *p* < 0.01) with respect to the aPM_2.5_ group. In addition, expression of these cytokine genes was also determined to further characterize the in vivo inflammatory response. Expression of *TNF-α* and *IL-1β* mRNA were increased 2.27- (^#^
*p* < 0.05) and 2.09-fold (^#^
*p* < 0.05) in the lungs of the aPM_2.5_-exposed rats at 11 weeks compared to the control group. The *IL-10* mRNA expression was not affected by aPM_2.5_ (Figure 3B). Protein and mRNA expression of IL-10 were increased in *Colla corii asini*-treated groups but the changes were not statistically significant. These results suggest that *Colla corii asini* treatment suppressed pro-inflammatory cytokine production in the aPM_2.5_ model.

### 2.5. Colla corii asini Inhibits Pathologic Changes and Lung Injury in Rats

Compared with control group lungs, intratracheal instillation of aPM_2.5_ resulted in interstitial/alveolar wall thickening and emphysematous changes in lung sections from rats after aPM_2.5_ exposure in week 11, as shown in Figure 4. The rats receiving aPM_2.5_ developed inflammatory cell infiltration, characterized by neutrophil and lymphocyte infiltration, which was, however, suppressed by 5 g/kg body weight of *Colla corii asini*. The pathologic damage was milder in the aPM_2.5_ plus *Colla corii asini* groups than in the aPM_2.5_ group, with less thickness of local alveolar septae and decreased macrophages infiltration. The results indicated that *Colla corii asini* alleviated, to some extent, lung injury induced by intratracheal instillation of aPM_2.5_ for 11 weeks in rats.

### 2.6. Role of Colla corii asini in Protection against aPM_2.5_-Induced Alteration of Amino Metabolic Profile

A number of potential biomarkers were identified and are listed in the Appendix A. From the score plots of the Principal Component Analysis (PCA) model (Appendix A) clear separation among the control, aPM_2.5_ and *Colla corii asini* groups was observed in lung tissue from rats subjected to intratracheal instillation of aPM_2.5_ in week 11. In this study, a total of six discriminating metabolites contributed to the classification of the control and aPM_2.5_ groups (Figure 5, Appendix A). Comparison of the levels of identified biomarkers in the aPM_2.5_ and *Colla corii asini* groups revealed that four of them were reversed by *Colla corii asini*, as shown in Figure 6. An additional 24 metabolites were also altered to different degrees (Appendix A). All of the discriminating biomarkers belonged to the identified metabolic pathways. *Colla corii asini* displayed an obvious protective effect by adjusting the disturbed metabolic pathways, including arginine and proline metabolism, aminoacyl-tRNA biosynthesis and nitrogen metabolism by searching the KEGG database.

Expression levels of *Arg-1* and *iNOS* mRNA were determined in the lungs of aPM_2.5_-exposed rats after 11 weeks to verify their role in the arginine metabolism pathway. The *Arg-1* mRNA level was increased 1.87-fold (^#^
*p* < 0.05) compared to the control group, which was reversed to some extent in the *Colla corii asini*-treated groups but the changes were not statistically significant. *Colla corii asini* treatment also reversed the elevated level of *iNOS* induced by aPM_2.5_ (Figure 7).

## 3. Discussion

A large amount of literature evidence supports a robust association between exposure to PM_2.5_ and pulmonary disease morbidity and mortality, therefore PM_2.5_-induced respiratory system damage has become a serious public health concern [19,20,21]. Our previous study successfully created a multi-component aPM_2.5_, based on the size and composition of actual PM_2.5_ collected in Beijing and demonstrated that inhalation of aPM_2.5_ by rats resulted in lung function decline and oxidative stress injuries [22]. In order to further explore the mechanisms of PM_2.5_-induced lung toxicity, we created an intratracheal instillation-injury animal model using these artificial particles and evaluated intervention with *Colla corii asini*, which has long been used in traditional Asian medicine to treat various ailments, including pulmonary dysfunction [17,23,24].

Our results showed that aPM_2.5_-induced lung injuries were associated with decreased lung function and linked to inflammation according to a series of biochemical parameters. The results were generally consistent with those from studies reported in the literature [25,26]. We observed that breath flow and volume parameters (tidal volume, expiratory volume, minute ventilation volume, expiratory flow at 50% tidal volume, peak expiratory flow, peak inspiratory flow) increased, while time parameters (relaxation time, expiratory time, inspiratory time) decreased, in animals after exposure to aPM_2.5_, indicating that the rats were in a state of shortness of breath [27]. The reason for the significant change in the respiratory function of the control group may be due to the irritative damage to the airway caused by the infusion of physiological saline. The results suggested that after intratracheal instillation of aPM_2.5_, pulmonary ventilation and gas storage function were decreased and airflow was blocked, which might result from pathologic changes such as alveolar height dilatation, alveolar wall rupture and fusion [28].

Published data suggest that ambient PM pollution causes oxidative stress [29] and may influence allergic [30], immunologic [31] and systemic inflammatory responses [32,33,34]. The results demonstrated that inflammation caused by the administration of aPM_2.5_ led to a substantial increase in the levels of WBC, LYM, MON, NEUT and EOS compared to the controls. To further explore the possible mechanisms of aPM_2.5_-induced pulmonary inflammation, we investigated the expression of inflammation-related cytokines in vivo at molecular and genetic levels. We observed an increase of pro-inflammatory cytokines (TNF-α, IL-1β) and a decrease of anti-inflammatory cytokine (IL-10), which were consistent with other published studies [35,36,37]. In addition, histological examination found inflammatory cell infiltration in the lung tissue.

A metabolomics approach was used to explore novel biomarker metabolites in the lung tissue from rats. This study reports for the first time that O-phospho-L-threonine may play a role in lung damage. Hypotaurine is known to modify the indices of oxidative stress and membrane damage, which is associated with lung adenocarcinoma [38,39,40]. Pipecolic acid (Pip), a non-protein lysine catabolite, is an important regulator of immunity in animals and plants [41,42,43]. In our study, the aPM_2.5_ group was characterized by increased levels of O-phospho-L-threonine, H-homoarginine, hypotaurine and pipecolic acid and decreased levels of 4-hydroxy-pyrrolidine-2-carboxylic and L-arginine. Our results demonstrate that arginine metabolism is the most indicative pathway after particle exposure for 11 weeks. Studies have shown that arginase and iNOS may regulate diverse pathways, including production of nitric oxide, polyamines and proline, which are critical components of immune suppression pathways [44,45,46]. Expression of arginase-1 is commonly believed to promote inflammation, fibrosis and wound healing [47]. *Arg-1* function was inhibited by aPM_2.5_ in our rat model, which might have contributed to the pathogenesis of lung injury [48]. The level of *iNOS* was increased in the aPM_2.5_ group compared to the control but this was not statistically significant. The roles of 4-hydroxy-pyrrolidine-2-carboxylic and H-homoarginine in arginine and proline metabolism still need further research. Studies have shown that L-arginine and its downstream metabolites could modulate innate and adaptive immunity to promote tumor survival and growth [49,50]. Nina E. King et al. have shown that asthmatic responses involve transport and metabolism of arginine by arginase in allergic airway inflammation. Scott J. et al. demonstrated that impairment of the L-arginine metabolism pathway can contribute to exacerbation of airway responsiveness to contractile agents [51]. Taking these observations and our findings together, it can be argued that altered arginine metabolism may represent a broad marker of lung damage and might be an early diagnostic marker of pulmonary disease.

In recent years, numerous traditional Chinese medicines have been found to possess potent pulmonary-protective activities and have attracted considerable interest as potential candidates for the development of novel therapies [17,24,52,53]. Our animal model revealed that *Colla corii asini* (1, 2, 5 g/kg/d, oral, for 11 weeks) inhibited aPM_2.5_-induced inflammation, reduced infiltration, maintained alveoli structure, inhibited pro-inflammatory cytokines (TNF-α and IL-1β) and elevated anti-inflammatory cytokine (IL-10) production in the lung. The number of CD8^+^ cytotoxic T cells in the lung correlates with the degree of airflow obstruction, suggesting that these cells cause tissue injury in lung. *Colla corii asini* decreased CD8+ level in the presence of aPM_2.5_ and also reversed the imbalance of CD4^+^/ CD8^+^ subsets in rats with airway inflammation. The defense mechanisms might involve the regulation of arginine and nitrogen metabolism pathways, as well as aminoacyl-tRNA biosynthesis. With the exception of H-homoarginine and pipecolic acid, changes in the levels of the other four identified metabolites were reversed by *Colla corii asini*. There were 24 additional discriminating metabolites involved in the protective effect of *Colla corii asini* against aPM_2.5_ injury. According to Michaela Kendall et al. [54], fine airborne urban particles (PM_2.5_) can sequester serine, asparagine and leucine following immersion in dilute human lavage fluid for short periods and suggested that this may involve lung opsonins, which is consistent with our findings. However, their role in protection against aPM_2.5_-induced alteration of amino metabolic pathways still needs further research. Our results showed that *Colla corii asini* significantly protected lung function and pathologic changes in response to aPM_2.5_. Our next study will focus on whether *Colla cori asini* could induce macrophage polarization towards a resolving M2 macrophage phenotype against lung injury.

In summary, our findings suggest that *Colla corii asini* could ameliorate lung function decline, suppress inflammation and regulate the balance of T lymphocyte subsets via suppression of arginase-1 to adjust the perturbed metabolic pathways induced by aPM_2.5_. Combining metabolomics data with pharmacological assays, we were able to obtain knowledge of the mechanisms of PM_2.5_-induced lung toxicity and the protective effects of *Colla corii asini* at an holistic level. Additionally, our work confirmed the utility of metabolomics platforms to dissect the underlying efficacy and protective mechanisms of traditional Chinese medicines.

## 4. Materials and Methods

### 4.1. Chemical Reagents

aPM_2.5_ samples were provided by Dr. QiMing Wang and obtained from Professor Zhonggao Gao Laboratories (Beijing, China). Samples were produced in accordance with a previous publication [22] using predominantly mesoporous silica nanoparticles (MSN), nitrate (NO3^−^), sulfate (SO_4_^2+^) and chromium (Cr). *Colla corii asini* (Lot: 20150701) was supplied by Shan Dong Dongeeiao Co., Ltd. (Shandong, China).

### 4.2. Animal Handling

Forty male (weighing 180–220 g) specific pathogen-free Sprague–Dawley (SD) rats, aged 6–8 weeks, were obtained from Beijing Vital River Laboratory Animal Technological Company (Beijing, China). All animal studies were conducted at the New Drug Safety Evaluation Center, Institute of Materia Medica, Chinese Academy of Medical Sciences and Peking Union Medical College. Rats were individually housed in suspended steel cages and acclimated in environmentally-controlled rooms (temperature of 20–26 °C; relative humidity of 40–70%; 12 h light/dark cycle) for 2 weeks prior to initiation of dosing, with food and water available *ad libitum*. The experiment protocol were approved by the laboratory animal welfare ethics committee of Beijing Union-Genius Pharmaceutical Technology Development Co., Ltd. (Beijing, China) (No.0000254, Aug. 16, 2016). The facilities achieved the Association for Assessment and Accreditation of Laboratory Animal Care International (AAALAC International) accreditation (1438#) and the animals were maintained according to the ‘Guide for the Care and Use of Laboratory Animals (Guide), NRC 2011.’

### 4.3. Rat model of Intratracheal Instillation of aPM_2.5_ and the Administration of Colla corii asini 

Dosing formulations of *Colla corii asini* were dissolved in pure deionized water, protected from light and stored at 2–8 °C until administration. aPM_2.5_ was dissolved in physiological saline, ultrasonically broken in a Sonifier cell disruptor and stored at 2–8 °C until administration. SD rats were randomly assigned to the following five groups (8 per group): (Ι) control group, (II) aPM_2.5_ group, (III –V) aPM_2.5_ + *Colla corii asini* groups (low-, mid- and high-dose groups). Rats in the aPM_2.5_ + *Colla corii asini* groups received *Colla corii asini* daily by oral gavage for 11 weeks at doses of 1, 2 and 5 g/kg body weight, whereas rats in the other two groups were treated with distilled water (10 mL/kg). Twice weekly intratracheal instillation of aPM_2.5_ (30 mg/kg, 1 mL/kg body weight) was used to induce lung injury, using a volume of 1 ml (kg·bw)^−1^, while the control group received equivalent volumes of physiological saline [55].

### 4.4. Pulmonary Function Measurement

Respiratory function was measured using a whole body plethysmography system (Emka Technologies, Paris, France) and the raw data were converted using specialized analysis software (IOX 2.9.4.32, Emka Technologies). Briefly, conscious rats were placed in plethysmography chambers for 10–15 min, eliminating data from animals adapting to their environments or sleeping. The test parameters, including respiratory rate (f), expiratory flow at 50% tidal volume (EF50), peak expiratory flow (PEF), peak inspiratory flow (PIF), tidal volume (TV), minute ventilation volume (MV), expiratory volume (EV), inspiratory time (Ti), relaxation time (RT) and expiratory time (Te), were recorded during a 3–8 min period once a week for 11 weeks [56].

### 4.5. T Lymphocyte Phenotype Analysis of Whole Blood

Fresh whole blood (100 µL in EDTA) was incubated with 5 µL of each antibody for 20 min in the dark, treated with 2 mL ACK Lysis Buffer (Sigma-Aldrich, St. Louis, MO, USA) and then centrifuged at 180 g for 5 min. After two washes with PBS/1% BSA (Sigma-Aldrich), the cells were resuspended in 2 mL PBS and analyzed immediately via flow cytometry using a FACSCalibur™ system with CellQuest™ software (BD Biosciences, Franklin Lakes, NJ, USA). All T cell samples were stained with APC conjugated mouse anti-rat CD3 antibody (BD Pharmingen, Franklin Lakes, NJ, USA). To determine the CD4^+^/CD8^+^ ratio, cells were stained with FITC conjugated mouse anti-rat CD8 (BD Pharmingen) and PE conjugated mouse anti-rat CD4 (BD Pharmingen) in the same test tube [57,58].

### 4.6. Bronchoalveolar Lavage Fluid Collection and Analysis

Bronchoalveolar lavage fluid (BALF) was obtained by slow perfusion of cold saline (12 mL) into the left lung three times via tracheal intubation [59]. The BALF was separated by centrifugation at 300 g for 10 min at 4 °C. The cell-free supernatants were assayed for cytokines. Assay kits for TNF-α, IL-1β and IL-10 were provided by Bio-Plex ProTM (Hercules, CA, USA). The cytokine levels were measured according to the manufacturer’s instructions. The cell pellets were resuspended in WBC dilution buffer (200 μL) for cell counts and classification. The following parameters were measured using a MEK-7222 K Automated Hematology Analyzer (Nihon-Kodhen Co., Tokyo, Japan): white blood cell (WBC), lymphocyte (LYM), monocyte (MON), neutrophil (NEUT), eosinophil (EOS) and basophil (BAS).

### 4.7. Lung Histopathology Analysis

Following BALF collection, lung right lobe specimens were removed and fixed with 10% neutral formaldehyde. Tissue blocks were embedded in paraffin and cut into 4-µm sections using an RM2145 microtome (LEICA, Wetzlar, Germany). The sections were stained with hematoxylin and eosin (H&E) and observed under a light microscope (LEICA).

### 4.8. LC-MS/MS-Based Metabolomics Analysis

Lung tissues (~50 mg) were prepared by mixing lung tissue homogenate (100 µL) with acetonitrile containing 0.1 mM norvaline (400 µL). After centrifugation at 15,000 g for 10 min, a 5 µL aliquot of the supernatant was injected into a LC-MS/MS system. The separation of lung components was performed on an ACQUITY UPLC BEH C18 column (3.5 m, 2.1 × 100 mm, Waters Corp, Milford, MA, USA). The mobile phase consisted of two solvents: mobile phase A [99.9% water and 0.1% aqueous formic acid (*v*/*v*) containing 10 mM ammonium acetate] and mobile phase B (49.95% acetonitrile, 49.95% isopropanol and 0.1% aqueous formic acid (*v*/*v*) containing 10 mM ammonium acetate). The running method was as described in a previous study [60]. The metabolites were quantified using an ACQUITY UPLC (Eksigent LC100) system coupled with a hybrid quadrupole time-of-flight mass spectrometer (Triple TOF 5600+, AB SCIEX). Metabolomics databases (Madison Metabolomics Consortium Database and METLIN) were searched to identify the structures of high-contribution score metabolites. MassLynx software (Waters Corp.) was used to acquire the chromatogram and mass spectrometric data in centroid format. The dataset was exported into SIMCA-P+ 12.0 (Umetrics, Kinnelon, NJ, USA) for further analysis after generation of a multivariate data matrix, as previously described [61].

### 4.9. Quantitative Reverse-Transcriptase DNA Polymerase Chain Reaction (qRT-PCR) Analysis

A TRIzol^®^ Plus RNA Purification Kit (Invitrogen, Hillsboro, OR, USA), following the manufacturer′s instructions, was used to extract total RNA from the right lung tissues of SD rats. Total RNA (0.5 μg) was reverse transcribed into cDNA using a HiScript II Q RT SuperMix IIa kit (Vazyme, Nanjing, China). PCR was carried out on a LightCycler^®^ 480 II Real-time PCR Instrument (Roche, Switzerland) with QuantiFast^®^ SYBR^®^ Green PCR Master Mix (Qiagen, Germany). The primer sequences were designed in the laboratory and synthesized by Generay Biotech (Generay, Shanghai, China) based on the mRNA sequences obtained from the NCBI database as described in Table 2. The mRNA expression levels were normalized to β-actin (ACTB) and were calculated using the 2^−ΔΔ*C*t^ method [52].

### 4.10. Statistical Analysis

The experimental data are presented as mean ± standard deviation (SD). SPSS 22.0 and GraphPad Prism 6.0 were used for statistical analysis and graph generation. Shapiro-Wilk test was first used to verify the normality of data. Comparisons between two groups were performed using a two-tailed unpaired Student’s t test. One-way analysis of variance (ANOVA) followed by Tukey′s post hoc tests were used to compare more than two populations. Significant differences were attributed to values with * *p* < 0.05, ** *p* < 0.01 and *** *p* < 0.001.

## Figures and Tables

**Figure 1 ijms-20-00055-f001:**
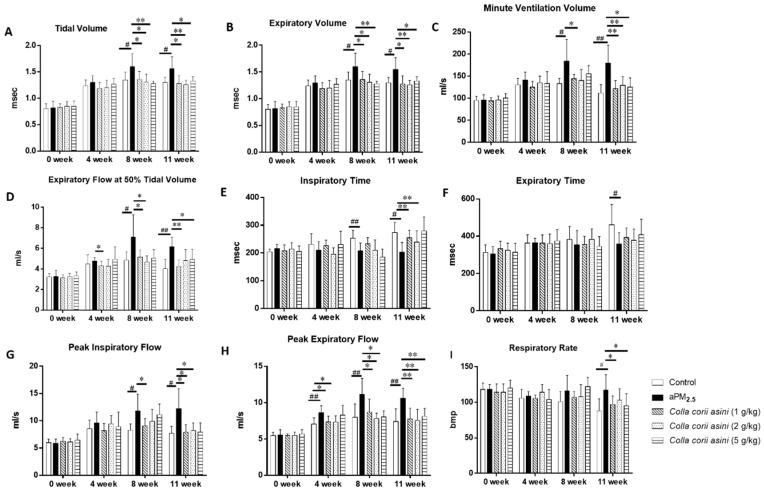
Protective effect of *Colla corii asini* in rats subjected to intratracheal instillation of aPM_2.5_. (**A**–**I**) Respiratory function changes of rats during intratracheal instillation of aPM_2.5_ for 11 weeks. (**A**) Tidal volume and (**B**). Expiratory volume (msec). (**C**). Minute ventilation volume. (**D**). Expiratory flow at 50% tidal volume (mL). (**E**). Inspiratory time and (**F**). Expiratory time (msec). (**G**). Peak inspiratory flow. (**H**). Peak expiratory flow (mL/s). (**I**). Respiratory rate (bmp). ^#^
*p* < 0.05 and ^##^
*p* < 0.01, versus the control group; * *p* < 0.05 and ** *p* < 0.01, versus the aPM_2.5_ group, *n* = 8.

**Figure 2 ijms-20-00055-f002:**
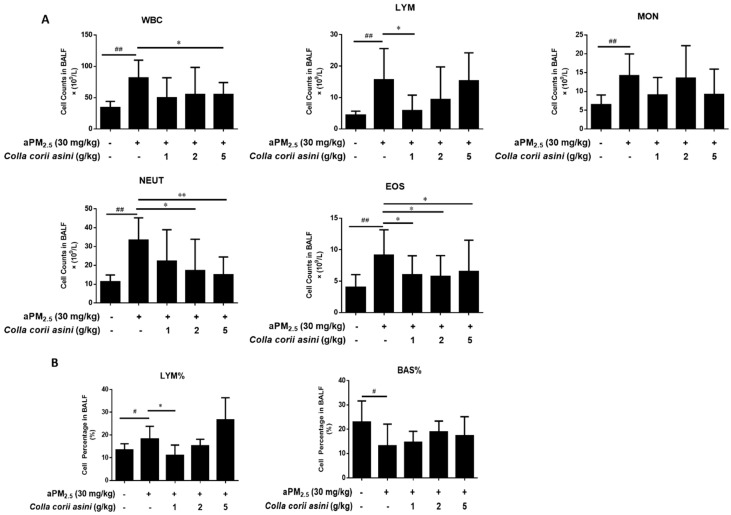
*Colla corii asini* prevented aPM_2.5_-induced inflammation. Effects of *Colla corii asini* on (**A**). cell counts (× 10^9^/L) and (**B**). cell percentage in the BALF of rats subjected to intratracheal instillation of aPM_2.5_ for 11 weeks. Data are expressed as mean ± SD (*n* = 8). ^#^
*p* < 0.05 and ^##^
*p* < 0.01, versus the control group; * *p* < 0.05 and ** *p* < 0.01, versus the aPM_2.5_ group.

**Figure 3 ijms-20-00055-f003:**
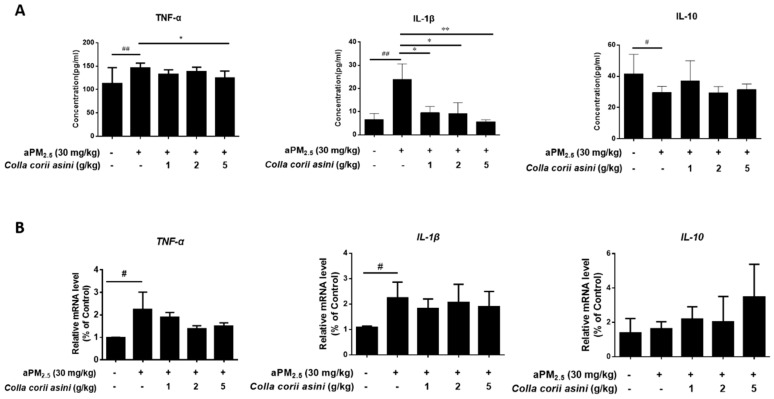
*Colla corii asini* prevented aPM_2.5_-induced inflammation. Effect of *Colla corii asini* on (**A**). TNF-α, IL-1β, IL-10 concentrations (pg/mL) in BALF (mean ± SD, *n* = 8) and (**B**). relative gene expression of *TNF-α*, *IL-1β* and *IL-10* in the lungs of rats subjected to intratracheal instillation of aPM_2.5_ for 11 weeks. Values are reported using the 2^−ΔΔ*C*t^ method and were obtained by comparison to those of the control group and normalized to β-actin (ACTB) (mean ± SD, *n* = 3). ^#^
*p* < 0.05 and ^##^
*p* < 0.01, versus the control group; * *p* < 0.05 and ** *p* < 0.01, versus the aPM_2__.5_ group.

**Figure 4 ijms-20-00055-f004:**
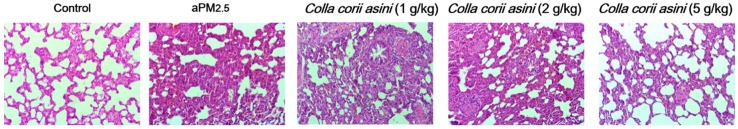
*Colla corii asini* prevented aPM_2.5_-induced inflammation. H&E-stained (×200) lung sections from rats after aPM_2.5_ exposure for 11 weeks. Histological examination and representative images are shown.

**Figure 5 ijms-20-00055-f005:**
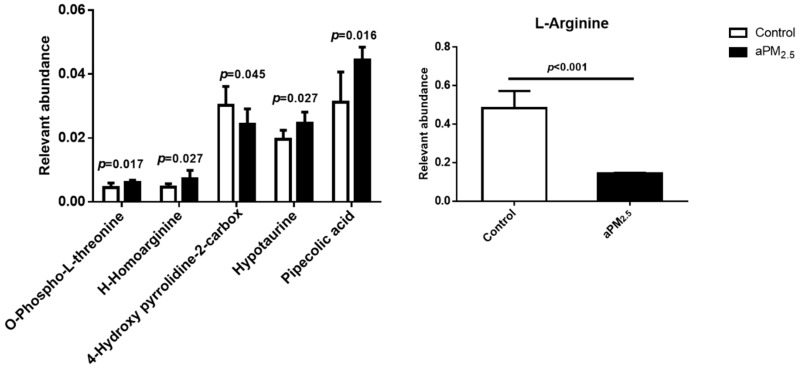
The discriminating metabolites obtained from lung tissue samples in the control and aPM_2.5_ groups by LC-MS/MS analysis. Data were calculated from the ratio of the mean peak areas in the control and aPM_2.5_ groups and normalized to L-norvaline (internal standard); *p* value of independent *t*-test, *n* = 8.

**Figure 6 ijms-20-00055-f006:**
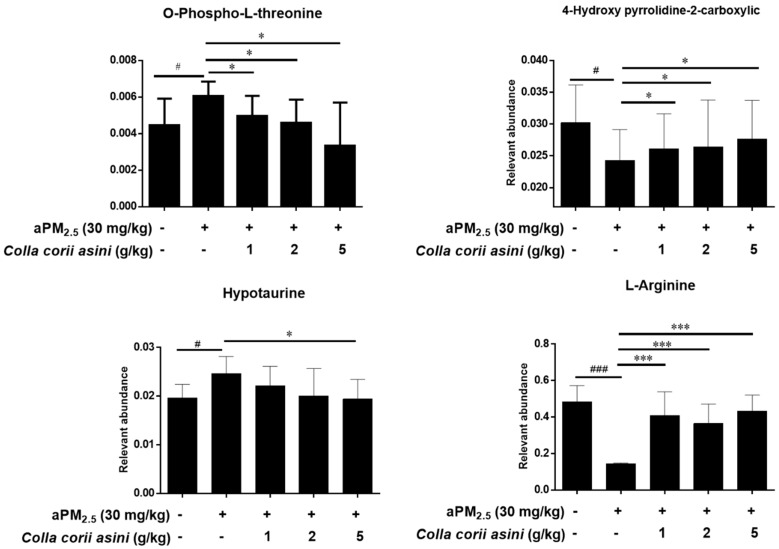
Protective effects of *Colla corii asini* against aPM_2.5_-induced alteration of amino metabolic profile. Data were calculated from the ratio of the mean peak areas in the aPM_2.5_ and *Colla corii asini* groups and normalized to L-norvaline (internal standard); ^#^
*p* < 0.05 and ^###^
*p* < 0.001, versus the control group; * *p* < 0.05 and *** *p* < 0.001, versus the aPM_2.5_ group, *n* = 8.

**Figure 7 ijms-20-00055-f007:**
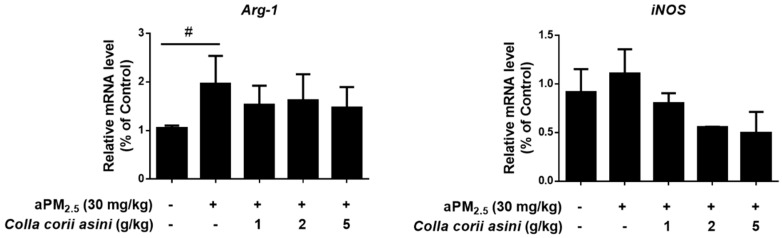
Effects of *Colla corii asini* on the relative gene expression of *Arg-1* and *iNOS* in the lungs of rats subjected to intratracheal instillation of aPM_2.5_ for 11 weeks. Values are reported using the 2^−ΔΔ*C*t^ method and were obtained by comparing with those of the control group and normalized to ACTB (mean ± SD, *n* = 3). ^#^
*p* < 0.05, versus the control group.

**Table 1 ijms-20-00055-t001:** Role of *Colla corii asini* in aPM_2.5_-induced alteration of T lymphocyte subsets.

Group	*Colla corii asini* (g/kg)	Number	CD3^+^ (%)	CD4^+^ (%)	CD8^+^ (%)	CD4^+^/CD8^+^
Control	-	8	55.29 ± 9.46	44.63 ± 8.89	10.26 ± 3.31	4.84 ± 2.06
aPM_2.5_	-	8	54.69 ± 5.52	44.44 ± 5.50	10.16 ± 4.56	5.54 ± 3.09
Low-dose group	1	8	51.28 ± 5.26	44.36 ± 5.49	6.18 ± 2.55 ^#,^*	8.48 ± 3.70 ^#^
Mid-dose group	2	8	58.44 ± 8.44	48.14 ± 10.1	7.01 ± 4.83	8.22 ± 3.47 ^#^
High-dose group	5	8	54.18 ± 4.79	47.07 ± 5.08	6.79 ± 2.93	8.95 ± 5.98 ^#^

Data are presented as the mean ± SD. ^#^
*p* < 0.05, significant difference between *Colla corii asini*-treated groups and control group; * *p* < 0.05, significant difference between *Colla corii asini*-treated groups and aPM_2.5_-treated group.

**Table 2 ijms-20-00055-t002:** Sequences of primers used for qRT-PCR.

Gene Name	Forward Primer	Reverse Primer
*TNF-α*	ACCAGCAGATGGGCTGTA	CTCCTGGTATGAAATGGCAA
*IL-1β*	CAAGGAGAGACAAGCAACG	TTCATCACACAGGACAGGTAT
*IL-10*	ACTGGACTGCAGGACATA	TTTGGAGAGAGGTACAAACGA
*iNOS*	TTGCTTCTGTGCTAATGCG	TCCGACTTTCCTGTCTCAGTA
*Arg-1*	TCGTACTGTGAACACGGC	GGTAGTCAGTCTCTGGCTTAT
*ACTB*	CCACCATGTACCCAGGCATT	CGGACTCATCGTACTCCTGC

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
