# Peer review of "Protective Effect of Colla corii asini against Lung Injuries Induced by Intratracheal Instillation of Artificial Fine Particles in Rats"

_ijms, 2018, doi:10.3390/ijms20010055_

Round 1
Reviewer 1 Report
Environmental issues pose huge threats to public health, particularly the damage caused by fine particulate matter PM2.5. The mechanisms of injury require further investigation and medical materials that can protect the lungs from PM2.5 are needed. The authors have found that the traditional Chinese medicine Colla corii asini may play a role in ameliorating PM2.5-induced lung toxicity. A rat model of lung injury was used to characterize the mechanisms of PM2.5-induced lung toxicity during a period of 11 weeks. Authors assess the potential protective effects of Colla corii asini by analysing pulmonary functioning, BALF cell count, lung histopathology images, inflammatory and anti-inflammatory proteins with their corresponding mRNAs, metabolites, and genes expression of Arg-1 and iNOS. Data was analysed by parametric tests i.e. t-test and one-way ANOVA with Tuckey post hoc test for multiple comparisons. The authors mainly focus on presenting differences between PM2.5 and PM2.5+ Colla corii asini and found that the medicine may protect against lung function decline, major pathologic changes, inflammation, and metabolite oxidative stress.
I congratulate the authors on the well written and organized manuscript. While the work is impressive, I have some concerns with respect to the statistical tests used and some of the conclusions drawn based on the results.
Below are some issues that need addressed:
Introduction:
1. Please provide a reference in line 49-50 (“Zhe showed that Colla corrii asini…”).
Results:
1. Results are difficult to follow without some introductory description about the animals and treatments used. Please add some of the information from lines 259-278 at the beginning of Section 2.1. Also, for every type of analysis, please clarify which samples have been used (e.g. serum week 1 to 11, or tissue week 11, etc.).
2. I have an overall concern about the statistical tests used, since data seems NOT normally distributed in several measurements presented in Figure 1, 2, 3, 5, 6, and 8 and Table 1, and authors do not mention checking for data distribution. For the presented analyses, a simple one-way ANOVA or a t-test was used, which are both prone to introduce errors when used in comparing too many groups alone. A better way to present this type of data is through a two-way repeated measurement ANOVA (if data is normally distributed) since you have 5 groups and a time line, and to focus on the interaction between time changes and treatment received in comparison with the non-treated (control) group. Finally, supplementary table S1 and S1 (no S2?) present no information about the concentrations/mass/a.u. (e.g. mean and 95%C.L. or mean±SD or median and IQR) which makes it difficult to follow. Please provide this additional information in the tables.
3. There is neither explanation nor discussion about why the respiratory functions in the control group change significantly during the experimental period (Figure 1).
4. Line 132: which protocol are you referring to?
5. Providing a scores plot of a supervised OPLS-DA analysis that force samples to separate will most of the time result in some separation, and hence, the results in Figure S1+S2 are not surprizing. There is no information about the validation used in the OPLS-DA modeling, and no predicted or cross-validated values are provided; hence, Figure S1+S2 make little sense. If the identified metabolites are indeed biomarkers, they should show a clustering of samples in an unsupervised PCA. Please provide a PCA scores plot containing ALL 5 groups in one figure and their clustering trends (or lack of clustering) instead of Figure S1+S2. In this way you will also show whether Colla corii asini indeed provides protection against lung injury.
6. Figure 7 provides little information, other than some possible affected metabolite pathways. It is interesting to know how many metabolites have been found per pathway in order to know whether the mentioned reactions are indeed the important ones. Currently, there are approximately 70-80 different metabolites in the aminoacyl-tRNA biosynthesis, app.40 in the nitrogen metabolism, 77 in arginine and proline metabolism, 20 in the taurine and hypotaurine metabolism, and app.50 in the lysine degradation. Identifying 1-5 metabolites in every of these pathways is not necessarily enough to conclude that these pathways were deranged, especially when gene expression results (Figure 8) did not show an overwhelming differnece between PM2.5 and PM2.5+Colla corii asini groups. If Figure 7 is to be presented in the study, please provide a table with Match Status of measured metabolites per number of actual metabolites in the mentioned pathways, p-values or -log(p) and the pathway impact.
7. Table S1+S1(2) in supplementary should contain values as mentioned above and should be presented in one single table. Please provide the values for all mentioned metabolites for comparison, even though they are not statistically significant in one or several groups. Do you use a pool of lung tissue and serum in this analysis? It is not clear whether the metabolites were identified in tissue or serum, or both.
Discussion:
1. There are some abbreviated parameters in lines 189 and 190 that have not been explained before line 285-288.
2. In such complex studies, there surely must be some strength and especially limitations, however, the authors do not provide any. Please provide such information.
Once again, I congratulate you for the nice study and well-written manuscript.
Author Response
Dear Editors and Reviewers,
Thank you very much for giving us an opportunity to revise our manuscript entitled “Protective effect of Colla corii asini against lung injuries induced by intratracheal instillation of artificial fine particles in rats” with the ID “ijms-397282”. Those comments are all valuable and helpful for revising and improving our manuscript, as well as the important guiding to our following investigations. We have studied the comments carefully and have made corrections which we hope meet with your approval. All of revised portions are marked in blue in the paper.
The main corrections in the paper and the responds to the reviewer’s comments are depicted as follows.
We greatly appreciate the reviewers’ time and comments.
Sincerely yours,
Hongtao Jin, corresponding author
Tiantian Liu, first author
Comments and Suggestions for Authors
Environmental issues pose huge threats to public health, particularly the damage caused by fine particulate matter PM2.5. The mechanisms of injury require further investigation and medical materials that can protect the lungs from PM2.5 are needed. The authors have found that the traditional Chinese medicine Colla corii asini may play a role in ameliorating PM2.5-induced lung toxicity. A rat model of lung injury was used to characterize the mechanisms of PM2.5-induced lung toxicity during a period of 11 weeks. Authors assess the potential protective effects of Colla corii asini by analysing pulmonary functioning, BALF cell count, lung histopathology images, inflammatory and anti-inflammatory proteins with their corresponding mRNAs, metabolites, and genes expression of Arg-1 and iNOS. Data was analysed by parametric tests i.e. t-test and one-way ANOVA with Tuckey post hoc test for multiple comparisons. The authors mainly focus on presenting differences between PM2.5 and PM2.5+ Colla corii asini and found that the medicine may protect against lung function decline, major pathologic changes, inflammation, and metabolite oxidative stress.
I congratulate the authors on the well written and organized manuscript. While the work is impressive, I have some concerns with respect to the statistical tests used and some of the conclusions drawn based on the results.
Reply: Thanks a lot for your professional and precise advice, and we have carefully revised the manuscript according to your comments / suggestions. Changes have been marked in blue in revised paper.
Introduction:
1. Please provide a reference in line 49-50 (“Zhe showed that Colla corrii asini…”).
Reply and action: Many thanks and we have added a reference to the line 52 of Page 2:
“Zhang, Z.; Ma, Y.; Hu, J.; Lan, H.; Zhang, Y.; Yao, C., Reverse effect of Ejiao (colla corri asini) on imbalance of Th17/Treg subsets of mice with airway inflmtion. Shandong Medical Journal 2018, (6), 11-14”
Results:
1. Results are difficult to follow without some introductory description about the animals and treatments used. Please add some of the information from lines 259-278 at the beginning of Section 2.1. Also, for every type of analysis, please clarify which samples have been used (e.g. serum week 1 to 11, or tissue week 11, etc.).
Reply and action: As suggested, we have revised the manuscript by adding the statement “The rats were exposed to aPM2.5 (30 mg/kg, 1 mL/kg body weight) via intratracheal instillation except for the controls, while rats in the aPM2.5 + Colla corii asini groups received Colla corii asini daily by oral gavage for 11 weeks at doses of 1, 2 and 5 g/kg body weight. Respiratory function of all animals was measured weekly throughout the treatment for 11 weeks.” at the beginning of Section 2.1 and details about which samples have been used for every type of analysis have also been marked in blue in revised paper:
P3, line 67, “for 11 weeks” has been added.
P4, line 89, “in the fresh whole blood of rats of 11 weeks” has been added.
P4, line 89-90, “Colla corii asini decreased CD8+ level in the presence of aPM2.5.” has been added.
P4, line 90-91, “the Colla corii asini+ aPM2.5 groups.” has been added.
P4, line 98, “in BALF week 11samples” has been added.
P6, line 130-131, “lung sections from rats after aPM2.5 exposure in week 11” has been added.
P6, line 135, “macrophages” has been added.
P7, line 144, “lung tissue and serum from rats subjected to intratracheal instillation of aPM2.5 in week 11” has been added.
2. I have an overall concern about the statistical tests used, since data seems NOT normally distributed in several measurements presented in Figure 1, 2, 3, 5, 6, and 8 and Table 1, and authors do not mention checking for data distribution. For the presented analyses, a simple one-way ANOVA or a t-test was used, which are both prone to introduce errors when used in comparing too many groups alone. A better way to present this type of data is through a two-way repeated measurement ANOVA (if data is normally distributed) since you have 5 groups and a time line, and to focus on the interaction between time changes and treatment received in comparison with the non-treated (control) group. Finally, supplementary table S1 and S1 (no S2?) present no information about the concentrations/mass/a.u. (e.g. mean and 95%C.L. or mean±SD or median and IQR) which makes it difficult to follow. Please provide this additional information in the tables.
Reply: Thanks a lot for your professional advice. For analysis of statistical significance, we first tested whether there was variation among the groups using Shapiro-Wilk test. Although a time line was involved in our study, samples have been used for analysis in our study were week 11.The specific analysis comparisons were performed in different dose groups, and one-way ANOVA can present analyses. We also checked some literature to support our study:
Hematopoietic effects and mechanisms of Fufang ejiao jiang on radiotherapy and chemotherapy-induced myelosuppressed mice. J Ethnopharmacol 2014, 152, (3), 575-84.
Role of the lipid-regulated NF-κB/IL-6/STAT3 axis in alpha-naphthyl isothiocyanate-induced liver injury. Archives of toxicology 2017, 91, (5), 2235-2244.
Suppression of lung inflammation by the methanol extract of Spilanthes acmella Murray is related to differential regulation of NF-κB and Nrf2. Journal of ethnopharmacology 2018, 217, 89-97.
3. There is neither explanation nor discussion about why the respiratory functions in the control group change significantly during the experimental period (Figure 1).
Reply and action: Many thanks. We have revised the manuscript by adding the statement “The reason for the significant change in the respiratory function of the control group may be due to the irritative damage to the airway caused by the infusion of physiological saline” in discussion (P9, line 199-201).
4. Line 132: which protocol are you referring to?
Reply and action: Thank you for your advice. We have deleted this statement for better reading (P7, line 141).
5. Providing a scores plot of a supervised OPLS-DA analysis that force samples to separate will most of the time result in some separation, and hence, the results in Figure S1+S2 are not surprizing. There is no information about the validation used in the OPLS-DA modeling, and no predicted or cross-validated values are provided; hence, Figure S1+S2 make little sense. If the identified metabolites are indeed biomarkers, they should show a clustering of samples in an unsupervised PCA. Please provide a PCA scores plot containing ALL 5 groups in one figure and their clustering trends (or lack of clustering) instead of Figure S1+S2. In this way you will also show whether Colla corii asini indeed provides protection against lung injury.
Reply and Action: Many thanks for your precise advice. We have provided a PCA scores plot containing all five groups in one figure and deleted the original Figure S1+S2 in supplementary.
6. Figure 7 provides little information, other than some possible affected metabolite pathways. It is interesting to know how many metabolites have been found per pathway in order to know whether the mentioned reactions are indeed the important ones. Currently, there are approximately 70-80 different metabolites in the aminoacyl-tRNA biosynthesis, app.40 in the nitrogen metabolism, 77 in arginine and proline metabolism, 20 in the taurine and hypotaurine metabolism, and app.50 in the lysine degradation. Identifying 1-5 metabolites in every of these pathways is not necessarily enough to conclude that these pathways were deranged, especially when gene expression results (Figure 8) did not show an overwhelming differnece between PM2.5 and PM2.5+Colla corii asini groups. If Figure 7 is to be presented in the study, please provide a table with Match Status of measured metabolites per number of actual metabolites in the mentioned pathways, p-values or -log(p) and the pathway impact.
Reply and action: Special thanks to you for your good comments. As suggested, we have deleted Figure 7 and revised the manuscript and recommendation by removing some of the comments in results and discussion (p7, line 146-148; p8, line 165-172).
7. Table S1+S1 (2) in supplementary should contain values as mentioned above and should be presented in one single table. Please provide the values for all mentioned metabolites for comparison, even though they are not statistically significant in one or several groups. Do you use a pool of lung tissue and serum in this analysis? It is not clear whether the metabolites were identified in tissue or serum, or both.
Reply and action: Thanks a lot for your suggestions. We agree with the reviewer and have revised Table S1 in the supplementary part of the new version. The metabolites were identified in tissue. Method for the serum samples has been removed from the manuscript, as this parameter results was not and will not be in the manuscript. We apologize for not removing it in the first submission.
Discussion:
1. There are some abbreviated parameters in lines 189 and 190 that have not been explained before line 285-288.
Reply and Action: As suggested, we have explained these abbreviated parameters in line 196-198 of Page 9:
“volume parameters (tidal volume, expiratory volume, minute ventilation volume, expiratory flow at 50% tidal volume, peak expiratory flow, peak inspiratory flow) increased, while time parameters (relaxation time, expiratory time, inspiratory time) decreased”.
2. In such complex studies, there surely must be some strength and especially limitations, however, the authors do not provide any. Please provide such information.
Once again, I congratulate you for the nice study and well-written manuscript.
Reply: We really appreciate your advice. We have addressed multiple large issues mainly at a descriptive level, given the current interest in how ultrafine particulate matter causes injury to the lung the suppression effect of Colla cori asini in the animal model of aPM2.5 in the present study. Our next study will investigate whether Colla cori asini could induce macrophage polarization towards a resolving M2 macrophage phenotype against lung injury and the role of Arginine‑regulated NF‑κB/IL‑6 axis in the switch of M1 or M2 macrophage polarization. We will focus on these elements and explore them in substantially more detail. Thanks again for your good comments.
Action: We have added “Our next study will focus on whether Colla corri asini could induce macrophage polarization towards a resolving M2 macrophage phenotype against lung injury” in line 255-257 of Page 10.

Reviewer 2 Report
The authors provide a manuscript that identifies a protective effect of Colla corii asini on aPM2.5-induced lung injury model. The authors utilize a combination of biochemical and physiological analyses to clarify this effect. The work is done in a reasonably thorough manner using multiple approaches. Some issues detract from the manuscript in its present form.
1. In Materials and Methods and Results, please add exact information of which samples (e.g. serum, lung tissue, BALF, etc) have been used.
2. You described the possibility of Colla corii asini as therapeutic medicine. Regarding the concentration of Colla corii asini in your animal protocol, is one of those concentration is similar to that in human with usual medication?
3. In Table 1 and Discussion, aPM2.5 did not have any influences on CD8+ level in whole blood. Therefore, you can just indicate that Colla corii asini decreased CD8+ level in the presence of aPM2.5 in line 82. You need to compare Colla corii asini- aPM2.5- group and Colla corii asini+ aPM2.5- group when you clarify the effect of Colla corii asini on CD8+ level.
4. Regarding Table 2 and Discussion, there is no discussion about the relationship between lower CD8+ level in whole blood and the effect of Colla corii asini on aPM2.5-induced lung injury model.
5. From histological images in Figure 4, you concluded that aPM2.5 resulted in emphysematous changes. In general, patients with emphysema show low peak expiratory flow. Can pulmonary functions explain emphysematous changes in lung? Also, you may be able to show emphysematous changes quantitatively (e.g. mean linear intercept).
6. In line 215, you described that “Arg-1 function was inhibited by aPM2.5 in our rat model”. However, you showed that aPM2.5 increased Arg-1 mRNA level in Figure 8, and there is no data about Arg-1 “function”. You should show the evidence of it or refer to articles.
Author Response
Dear Editors and Reviewers,
Thank you very much for giving us an opportunity to revise our manuscript entitled “Protective effect of Colla corii asini against lung injuries induced by intratracheal instillation of artificial fine particles in rats” with the ID “ijms-397282”. Those comments are all valuable and helpful for revising and improving our manuscript, as well as the important guiding to our following investigations. We have studied the comments carefully and have made corrections which we hope meet with your approval. All of revised portions are marked in blue in the paper.
The main corrections in the paper and the responds to the reviewer’s comments are depicted as follows.
We greatly appreciate the reviewers’ time and comments.
Sincerely yours,
Hongtao Jin, corresponding author
Tiantian Liu, first author
Comments and Suggestions for Authors
The authors provide a manuscript that identifies a protective effect of Colla corii asini on aPM2.5-induced lung injury model. The authors utilize a combination of biochemical and physiological analyses to clarify this effect. The work is done in a reasonably thorough manner using multiple approaches. Some issues detract from the manuscript in its present form.
Reply: Many thanks for your advice, and we have carefully revised the manuscript according to your comments / suggestions. Changes have been marked in blue in revised paper.
1. In Materials and Methods and Results, please add exact information of which samples (e.g. serum, lung tissue, BALF, etc) have been used.
Reply and Action: As suggested, we have revised the manuscript by adding the following statement:
P3, line 67, “for 11 weeks” has been added.
P4, line 89, “in the fresh whole blood of rats of 11 weeks” has been added.
P4, line 89-90, “Colla corii asini decreased CD8+ level in the presence of aPM2.5.” has been added.
P4, line 90-91, “the Colla corii asini+ aPM2.5 groups.” has been added.
P4, line 98, “in BALF week 11samples” has been added.
P6, line 130-131, “lung sections from rats after aPM2.5 exposure in week 11” has been added.
P6, line 135, “macrophages” has been added.
P7, line 144, “lung tissue and serum from rats subjected to intratracheal instillation of aPM2.5 in week 11” has been added.
2. You described the possibility of Colla corii asini as therapeutic medicine. Regarding the concentration of Colla corii asini in your animal protocol, is one of those concentration is similar to that in human with usual medication?
Reply: Many thanks for your advice. According to the previous study, the maximum tolerated dose of Colla corii asini in ICR mice is more than 20.0g/kg(Maximum tolerable dose, and the clinical dose of Colla corii asini is relatively high, 3-9g/person one day,so the animal dosage is relatively high,maybe this is indeed a characteristic of pharmacological research of traditional Chinese medicine. The concentration in our study is comparable to that in human with usual medication.
3. In Table 1 and Discussion, aPM2.5 did not have any influences on CD8+ level in whole blood. Therefore, you can just indicate that Colla corii asini decreased CD8+ level in the presence of aPM2.5 in line 82. You need to compare Colla corii asini- aPM2.5- group and Colla corii asini+ aPM2.5- group when you clarify the effect of Colla corii asini on CD8+ level.
Reply and Action: We really appreciate your advice, and we have added more explanation in line 89-91 of page 4:” Colla corii asini decreased CD8+ level in the presence of aPM2.5. Compared with the aPM2.5 group, the CD4+/CD8+ ratio increased in the Colla corii asini+ aPM2.5 groups.” Many thanks.
4. Regarding Table 1 and Discussion, there is no discussion about the relationship between lower CD8+ level in whole blood and the effect of Colla corii asini on aPM2.5-induced lung injury model.
Reply and Action: Many thanks, we have added the following sentences to the line 242 of Page 10 in discussion:
“The number of CD8+ cytotoxic T cells in the lung correlates with the degree of airflow obstruction, suggesting that these cells may involving causing tissue injury in lung. Colla corii asini decreased CD8+ level in the presence of aPM2.5 and also reversed the imbalance of CD4+/ CD8+ subsets in rats with airway inflammation”
5. From histological images in Figure 4, you concluded that aPM2.5 resulted in emphysematous changes. In general, patients with emphysema show low peak expiratory flow. Can pulmonary functions explain emphysematous changes in lung? Also, you may be able to show emphysematous changes quantitatively (e.g. mean linear intercept).
Reply: Many thanks for your advice and your suggestions will be very valuable in our future research. We observed peak expiratory flow increased in rats after exposure to aPM2.5, indicating that the animals were in a state of shortness of breath. The results also suggested that after intratracheal instillation of aPM2.5, pulmonary ventilation and gas storage function were decreased, and airflow was blocked, which might result from pathologic changes such as alveolar height dilatation, alveolar wall rupture and fusion. The relationship between pulmonary functions and emphysematous changes in lung still needs further investigation, on animal models of lung injury of varying degrees and developing stage.
6. In line 215, you described that “Arg-1 function was inhibited by aPM2.5 in our rat model”. However, you showed that aPM2.5 increased Arg-1 mRNA level in Figure 8, and there is no data about Arg-1 “function”. You should show the evidence of it or refer to articles.
7. Reply and Action: Many thanks for your accurate advice. The gene Arg-1 are key regulators of critical processes associated airway tone, cell hyperplasia, and collagen deposition, respectively. Recent data suggest that arginase induction is not just a marker of allergic airway responses, but that arginase is involved in the pathogenesis of multiple aspects of disease. We cited literature related to our research and add the information to this manuscript. We hope what we have done can meet your requirements.

Round 2
Reviewer 1 Report
Thank you for taking my comments and suggestions into consideration.
What are the units in Table S1? Can you use whole numbers instead of sceintific Es? Maybe you can write uM(?) instead of M?
Reviewer 2 Report
Thank you for your revised manuscript and comments. You answered my questions well. I hope your success.